# Identifying and tailoring C−N coupling site for efficient urea synthesis over diatomic Fe−Ni catalyst

Xiaoran Zhang[1,2,6], Xiaorong Zhu [3,6], Shuowen Bo[4,6], Chen Chen [1] ✉, Mengyi Qiu[1], Xiaoxiao Wei[1], Nihan He[1], Chao Xie [1], Wei Chen[1], Jianyun Zheng [1] ✉, Pinsong Chen[2], San Ping Jiang [2,5] ✉, Yafei Li [3], Qinghua Liu [4] & Shuangyin Wang [1] ✉

Electrocatalytic urea synthesis emerged as the promising alternative of Haber−Bosch process and industrial urea synthetic protocol. Here, we report that a diatomic catalyst with bonded Fe−Ni pairs can significantly improve the efficiency of electrochemical urea synthesis. Compared with isolated diatomic and single-atom catalysts, the bonded Fe−Ni pairs act as the efficient sites for coordinated adsorption and activation of multiple reactants, enhancing the crucial C−N coupling thermodynamically and kinetically. The performance for urea synthesis up to an order of magnitude higher than those of single-atom and isolated diatomic electrocatalysts, a high urea yield rate of 20.2 mmol $h^{-1}$ $g^{-1}$ with corresponding Faradaic efficiency of 17.8% has been successfully achieved. A total Faradaic efficiency of about 100% for the formation of value-added urea, CO, and $NH_3$ was realized. This work presents an insight into synergistic catalysis towards sustainable urea synthesis via identifying and tailoring the atomic site configurations.

Urea is served as an indispensable chemical for fertilizer and a promising feedstock in fuel cell system[1,2]. In industry, urea is synthesized from liquid ammonia and carbon dioxide under the conditions of high pressure and elevated temperature[3,4]. However, the current urea synthetic protocol suffers from high energy input and the consumption of value-added ammonia, far away from meeting the demands for sustainable development[5]. Consequently, the electrocatalytic urea synthesis has emerged as one of the promising alternatives for the traditional urea manufacture to realize the urea production under milder conditions[6–11]. Compared with the indirect method of initial ammonia production with subsequent urea synthesis, the direct urea synthesis from co-activation and coupling of carbon dioxide ($CO_2$) and

nitrate ($NO_3^-$) waste plays the crucial role in the development of urea synthetic processes, to close the global carbon footprint, maintain nitrogen balance and reform the urea industry simultaneously[5,12,13].

The co-activation of carbon and nitrogen sources and electrocatalytic C−N coupling of in situ generated species guide the development direction of urea synthesis. Nevertheless, the formidable challenge is that the parallel $CO_2$ reduction reaction ($CO_2RR$) and $NO_3^-$ reduction reaction ($NO_3RR$), and inescapable hydrogen evolution reaction (HER) at negative potentials strongly compete with the desirable urea formation, resulting in complex products distribution and low efficiency of urea product[10,11]. The electrocatalytic urea synthesis consists of multistep electrochemical processes (proton-coupling

[1]State Key Laboratory of Chem/Bio-Sensing and Chemometrics, College of Chemistry and Chemical Engineering, Hunan University, Changsha, People's Republic of China. [2]WA School of Mines: Minerals, Energy & Chemical Engineering, Curtin University, Perth, WA 6102, Australia. [3]College of Chemistry and Materials Science, Nanjing Normal University, Nanjing, Jiangsu, People's Republic of China. [4]National Synchrotron Radiation Laboratory, University of Science and Technology of China, Hefei, People's Republic of China. [5]Foshan Xianhu Laboratory of the Advanced Energy Science and Technology Guangdong Laboratory, Foshan 528216, People's Republic of China. [6]These authors contributed equally: Xiaoran Zhang, Xiaorong Zhu, Shuowen Bo. ✉e-mail: chenc@hnu.edu.cn; jyzheng@hnu.edu.cn; s.jiang@curtin.edu.au; shuangyinwang@hnu.edu.cn

electron-transfer, PCET) and chemical steps (C−N coupling). To pursuit for efficient urea production, the above factors should be taken into account in catalyst design, which not only needs to meet the co-activation and reaction of reactants, but also needs to optimize the adsorption of intermediate species, construct efficient sites conducive to C−N coupling, and to reduce the occurrence of side reactions. The previous works mainly focused on the activities of sub-reactions with little attentions on the identification and understanding of C−N coupling sites[8–11]. Thus, it is urgent to explore the comprehensive strategies to meet the above demands and to identify the underlined reaction kinetics and mechanisms.

Herein, we designed the diatomic electrocatalyst to serve as the efficient electrocatalyst towards urea synthesis. The "three-in-one" of active site, activation site, and coupling site was realized in the bonded Fe−Ni pairs. Compared with the Fe-SAC and Ni-SAC, the simultaneous introduction of Fe and Ni sites into isolated diatomic Fe−Ni electrocatalyst (I-FeNi-DASC) overcomes the limitations of selective adsorption and activation of carbon-reactant or nitrogen-reactant unilaterally. The synergistic effect significantly improves the electrochemical urea synthesis via achieving the coordinated adsorption and activation of multiple reactants. The isolated Fe−$N_4$ and Ni−$N_4$ sites trigger numerous activated C- and N-species and increase possibility for the encounter and coupling of those inter-mediate species to generate crucial C−N bonds. More importantly, we found out that the diatomic electrocatalyst with bonded Fe−Ni pairs (B-FeNi-DASC) is the most effective for urea formation intrin-sically owing to the thermodynamic and kinetic feasible C−N cou-pling process on the bridge sites of Fe−Ni pairs (FeNi−$N_6$), and the effective suppressed HER on the bridged configuration. The per-formance for urea synthesis is up to an order of magnitude higher than those of Fe-SAC, Ni-SAC, and I-FeNi-DASC electrocatalysts, achieving a high urea yield rate of 20.2 mmol h$^{-1}$ g$^{-1}$ with corre-sponding Faradaic efficiency of 17.8%. Both the operando synchrotron-radiation Fourier transform infrared spectroscopy

(SR-FTIR) measurements and theoretical calculation indicate that the urea synthesis over B-FeNi-DASC arise from the coupling of *NH (the asterisk represents the adsorption site) and *CO to form the first C−N bond and the subsequent C−N coupling between *NHCO with *NO to form the second C−N bond, both are thermodynamic spon-taneous and highly kinetic feasible with low energy barriers of 0.21 and 0.09 eV on the bonded Fe−Ni pair site, respectively. This work presents new insights into identifying and tailoring activation sites and C−N coupling sites to boost the urea synthesis intrinsically, expected to promote the green revolution of urea industrial and provide guidance for catalytic coupling reactions.

## Results and discussion
### Physical characterization of electrocatalysts

The single atom catalysts (SACs) hold a great potential in electro-catalysis but encounter considerable challenges for more complicated catalytic reactions involving multiple reactants and intermediate spe-cies, thus the construction of diatomic sites emerged as the efficient strategy to overcome the inherent structural simplicity of the active centers in SACs[14–24]. Here, a series of single-atom and diatomic elec-trocatalysts decorated on nitrogen-doped carbon support were obtained via pyrolysis of the coordination polymer, as illustrated in Supplementary Fig. 1. The obtained catalysts were denoted as Fe-SAC, Ni-SAC, I-FeNi-DASC, and B-FeNi-DASC according to the site config-urations (see "Methods" for more details). These electrocatalysts dis-play uniform nanosphere microstructure, which was derived from the assembly of organic reactants as shown in the scanning electron microscopy (SEM) image (Supplementary Fig. 2). The elemental map-pings in Supplementary Fig. 3 indicate that Fe, Ni, N, and C elements are homogenously distributed over B-FeNi-DASC. According to the transmission electron microscope (TEM) and aberration-corrected high-angle annular dark-field scanning transmission electron micro-scopy (HAADF-STEM) images of B-FeNi-DASC (Fig. 1a, b), highly dis-persed bright dots were anchored on porous carbon matrix,

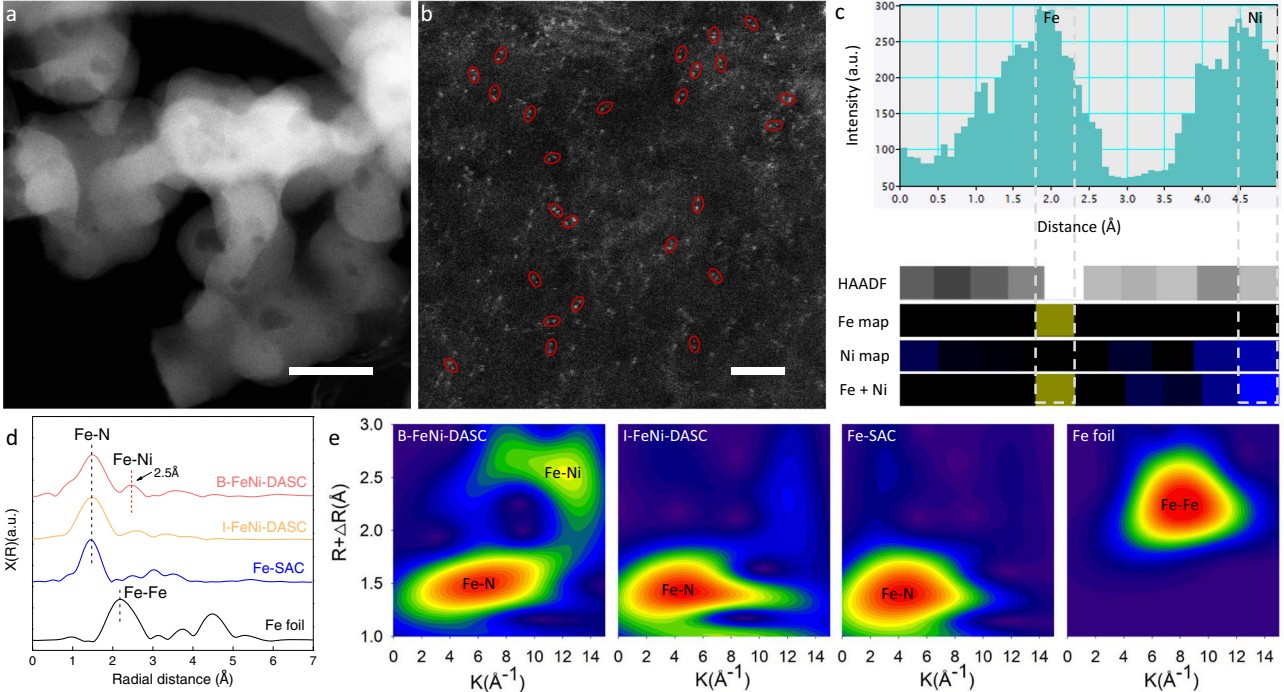

**Fig. 1 | Morphology and structure of single-atom and diatomic catalysts. a** TEM image and **b** Aberration-corrected HAADF-STEM image of B-FeNi-DASC. The scale bar is 100 and 2 nm, respectively. **c** Acquired HAADF-STEM image intensity profile accompanied by atomic-resolution EELS mapping of the Fe−Ni pair. **d** Fourier transform extended X-ray absorption fine structure (FT-EXAFS) spectra of Fe-SAC, I-FeNi-DASC, and B-FeNi-DASC. **e** Wavelet-transform plots for Fe element of Fe foil, Fe-SAC, I-FeNi-DASC, and B-FeNi-DASC.

suggesting the atomically dispersed structure of Fe and Ni sites[15–17]. With utilization of a host-guest strategy[21], the atomic pairs (marked by red circles) in B-FeNi-DASC imply the formation of bonded Fe−Ni dual sites, which is most likely originated from adsorption of Ni salts and the formed bonding with neighboring Fe atoms. Meanwhile, the atomic-resolution elemental analysis via electron energy-loss spectroscopy (EELS) line scan clearly confirms the existence of Fe−Ni pairs with a typical atomic distance of about 0.25 nm (Fig. 1c and Supplementary Fig. 4). By contrast, STEM images of Fe-SAC, Ni-SAC and I-FeNi-DASC were list in Supplementary Fig. 5.

The controlled synthesis of electrocatalysts with specific site configurations is further validated by X-ray absorption spectroscopy (XAS), and the X-ray absorption near edge structure (XANES) data at the Fe *K*-edge of Fe-SAC, I-FeNi-DASC, and B-FeNi-DASC clearly indicate that the peak position situates at higher energy than those of Fe foil (Supplementary Fig. 6), suggesting that the average valence state of Fe in Fe-SAC, I-FeNi-DASC and B-FeNi-DASC is higher than metallic $Fe^0$. Furthermore, the Fe K-edge of I-FeNi-DASC is lower than that of B-FeNi-DASC, implying the decreased valence of Fe in I-FeNi-DASC and the regulated coordination environment[23,24]. Moreover, the Fourier transform extended X-ray absorption fine structure (FT-EXAFS) analysis for Fe in Fe-SAC, I-FeNi-DASC, and B-Fe-Ni-DASC show the similar peak position at around 1.5 Å originated from Fe−N bonds (Fig. 1d), meanwhile, barely no metallic Fe−Fe bonds appear in those catalysts, corresponding to the fore mentioned results. In terms of extended X-ray absorption fine structure (EXAFS) of B-FeNi-DASC, the predominant peaks appeared in the first coordinated shells (1–2 Å) upon R space curve of Fe K-edge and Ni K-edge, which originate from the scattering of 1st shell Fe−N and Ni−N path, are almost the same in position (at ~1.5 Å) and magnitude, indicating nearly identical coordination environment for Fe and Ni atoms in the catalyst of B-FeNi-DASC. Notably, the broad peak appears upon the 2nd shell of Fe K-edge and Ni K-edge with a single-scattering path of at around 2.5 Å appeared in the second scattering shells (2–3.5 Å) (Fig. 1d, Supplementary Fig. 7 and Supplementary Table 1). This distance is in the range of the observed separation of dual-atom pairs in atomic resolution STEM imaging (Fig. 1c), which is consistent with the previous reports[25,26], therefore, we attributed this scattering path to the formation of Ni−Fe dual-atom pairs, in which a Fe atom connect to the Ni atom except for coordinate with 3N sites. Taken together, Ni−Fe diatomic configuration is formed in the B-FeNi-DASC. By contrast, for the catalyst of I-FeNi-DASC, the first coordinated shells for both Fe and Ni are similar to that of B-FeNi-DASC, but the peak in the 2nd shell of EXAFS R space is gentle, indicating that there are almost no Fe-Ni pairs exist in the catalyst of I-FeNi-DASC. On the other hand, compared to Fe-SAC and Ni-SAC, the 1st shell scattering (Fe−N and Ni−N) for B-FeNi-DASC displays asymmetry and slightly decreased magnitude, indicating that the chemical state of Fe is altered by the coupling Ni atom. Wavelet transform (WT)-EXAFS was also conducted to identify the metal-N and metal-metal paths in B-FeNi-DASC. This conclusion is clearly illustrated in the WT- EXAFS analysis (Fig. 1e). The EXAFS fittings reveal the Fe−$N_4$ and Ni−$N_4$ configurations of Fe and Ni in Fe-SAC, Ni-SAC (Supplementary Fig. 7 and Supplementary Table 1). However, the FeNi−$N_6$ configuration with the most consistent coordination environment to experimental fitting results has been evidently demonstrated as the optimized structure of the active unit in B-FeNi-DASC. The other physical characterization for comparing structure properties of B-FeNi-DASC, Fe-SAC, Ni-SAC, and I-FeNi-DASC are listed in Supplementary Figs. 8–12 and Supplementary Table 2. (Relevant descriptions are also listed in Supplementary Information).

## Realizing coordinated activation and promoting C−N coupling for urea synthesis over diatomic catalyst

The electrochemical measurements were carried out on a typical three-electrode configuration (Supplementary Fig. 13), and the linear sweep voltammetry (LSV) tests were performed initially to evaluate the electrochemical response of those electrocatalysts towards $CO_2$RR and $NO_3$RR, respectively. As shown in Supplementary Fig. 14, the Ni-SAC is highly active towards the $CO_2$RR with the low onset potential of about −0.25 V versus reversible hydrogen electrode (RHE). However, the current density remains almost unchanged when the nitrate ions was applied as the feedstock, indicating negligible activity to $NO_3$RR. On the contrary, the Fe-SAC shows a large gap of current density to catalyze $NO_3$RR than the reaction without nitrate ions but negligible response to the $CO_2$RR. The distinguishing activities originated from that the Ni sites in Ni-SAC are favorable for the absorption and activation of $CO_2$ but the Fe sites in Fe-SAC tend to be occupied by the nitrate reactant[27–30]. The subsequent product analysis of relevant reactions demonstrates that the Faradaic efficiency (FE) of $CO_2$RR to CO on Ni-SAC is up to 86.9% but a much lower FE of 19.9% was obtained for $NH_3$ generation at −1.5 V versus RHE. In the case of FE-SAC, the FE of $CO_2$RR to CO and $NO_3$RR to $NH_3$ is 19.2 and 65.2%, respectively, in contrast to that of Ni-SAC. The above results indicate that the reactions on Ni-SAC and Fe-SAC are dominated by $CO_2$RR or $NO_3$RR, respectively, inducing a unilateral scarcity of intermediate species and the low yield rates of urea, as well as undesirable efficiencies (Fig. 2b).

Above results indicate that owing to the structural simplicity of the active centers, the single-atom system possesses the intrinsic disadvantages to handle the complex catalytic reaction involving multiple reactants and intermediate species[31]. The construction of diatomic site configurations has emerged as the feasible strategy to realize the coordinated adsorption and activation, and efficient electrocatalytic coupling reaction. As illustrated in Supplementary Fig. 14, the I-FeNi-DASC with isolated Fe and Ni sites exhibits desirable response both to $CO_2$RR and $NO_3$RR, and the FE for $CO_2$RR to CO and $NO_3$RR to $NH_3$ is 82.5 and 78.9% at −1.5 V respectively (Fig. 2b), implying good activity for both reactions. Moreover, the electrocatalysts showed an improved urea yield rate of 10.7 mmol $h^{-1}$ $g^{-1}$ with a corresponding FE of 3.8%. On the other hand, the physically mixed Fe-SAC and Ni-SAC electrocatalyst (M-FeNi-DASC) exhibits the limited improvement and relatively lower performance than I-FeNi-DASC. Thus, the boosting of urea synthesis performance is mainly attributed to the synergistic effect in the constructed diatomic Fe−$N_4$ and Ni−$N_4$ sites[14,32]. The substitution of nitrite to nitrate as the feedstock barely affects the performance of urea synthesis as illustrated in Supplementary Fig. 15, suggesting that the constructed diatomic sites have the ability to catalyze the $NO_3$RR efficiently, since the nitrate reduction to nitrite was reported as the rate-determining step in the previous reports[33,34]. The direct coupling of carbon dioxide with nitrate (the relatively stable and abundant state) rather than mostly reported nitrite to achieve the efficient urea synthesis was realized in this work[8,10].

The design of isolated diatomic sites can trigger numerous activated C- and N-species simultaneously according to the product distributions in Fig. 2b, and provides the possibility for the encounter and chemical coupling to generate crucial C−N bonds. Nevertheless, it is not enough to provide a rich variety of intermediate species without the identification and construction of effective C−N coupling sites for efficient urea synthesis, and the further improvement of electrocatalytic performance remains unsolved. To this end, the introduction of bonded Fe−Ni pairs is expected to not only retains the synergistic effect in the I-FeNi-DASC but also enhance the reaction kinetics of C−N coupling process, to improve the electrochemical urea synthesis intrinsically. The B-FeNi-DASC exhibits comparable FE of 73.3% for $CO_2$RR to CO and 77.6% for $NO_3$RR to $NH_3$, respectively. More importantly, the B-FeNi-DASC possesses the superior urea synthetic ability among those of single-atom and diatomic electrocatalysts. The urea yield rate increases along with the applied negative potentials and delivers the highest urea yield rate of 20.2 mmol $h^{-1}$ $g^{-1}$ with a high FE of 17.8% at −1.5 V, superior to those of noble-metal based electrocatalysts[6–8].

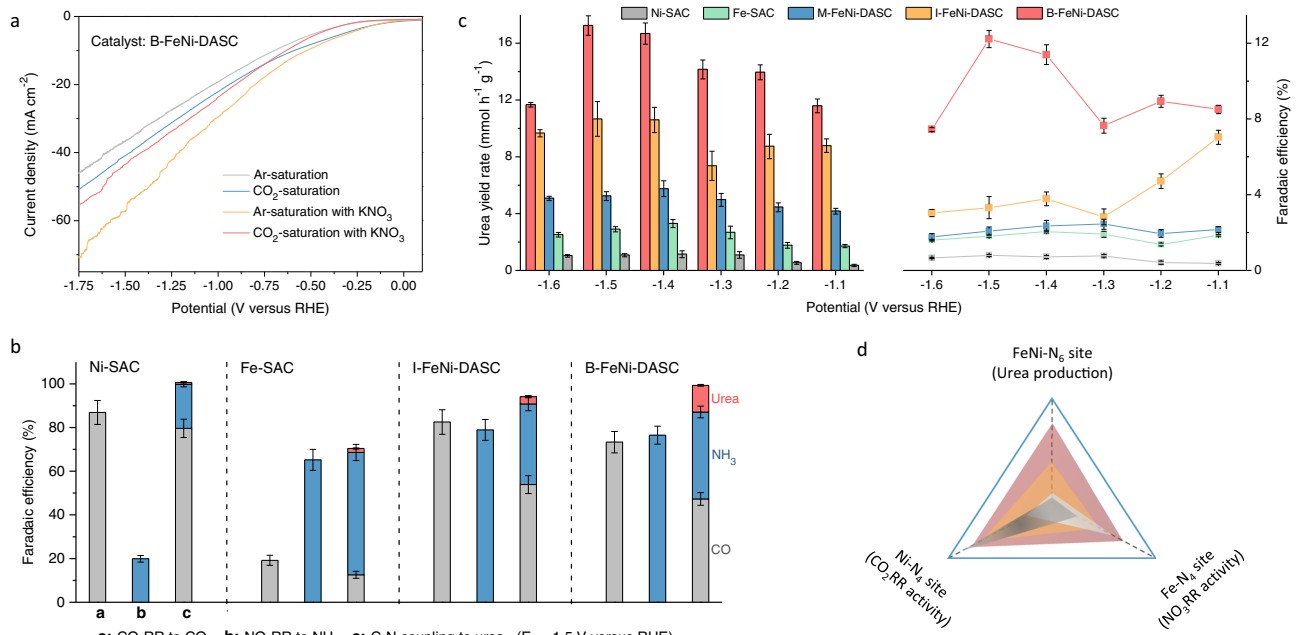

**Fig. 2 | Electrocatalytic performances for urea synthesis. a** LSV curves over B-FeNi-DASC. **b** The product distributions of $CO_2RR$, $NO_3RR$ and urea synthesis on Ni-SAC, Fe-SAC, I-FeNi-DASC, and B-FeNi-DASC at −1.4 V versus RHE. **c** Urea yield rates and corresponding Faradaic efficiencies on Ni-SAC, Fe-SAC, M-FeNi-DASC, I-FeNi-DASC, and B-FeNi-DASC at various applied potentials. **d** Illustrated correlation between $CO_2RR$ activity, $NO_3RR$ activity, and urea production over various site configurations. The error bars represent the standard deviation for at least three independent measurements.

The isotope labeling experiments were carried out to determine the nitrogen source in urea products. As illustrated in Supplementary Figs. 17 and 18, the $^1H$ NMR spectra identified the obtained $CO(^{15}NH_2)_2$-derived $^{15}NH_4^+$ according to a distinguishable chemical shift of triplet coupling of $^{14}N$ and doublet coupling of $^{15}N$, which corresponding to the calibration curves of authentic $CO(^{14}NH_2)_2$-derived $^{14}NH_4^+$ and $CO(^{15}NH_2)_2$-derived $^{15}NH_4^+$, substantially verifying the urea production from electrocatalytic coupling of $CO_2$ with $NO_3^-$ [35]. The urease decomposition-NMR method affords an indirect approach for identification and quantification of urea. On the other hand, the direct NMR spectra in Supplementary Figs. 17 and 19 with typical doublet coupling of $CO(^{15}NH_2)_2$ and single coupling of $CO(^{14}NH_2)_2$, which indeed derived from electrocatalytic coupling of $CO_2$ with $^{15}KNO_3/^{14}KNO_3$ respectively. Meanwhile, the total FE for the formation of value-added CO, $NH_3$ and urea products reaches about 100%, negligible hydrogen was produced in the co-electrolysis measurement over B-FeNi-DASC system, implying that the competing HER is almost completely suppressed. This is further validated by the LSV curves (Fig. 2a) and chronoamperometry curves (Supplementary Fig. 20)[36]. During ten continuous cycles, the urea yield rate and the current density (Supplementary Fig. 21) show no obvious decay, implying the high electrochemical durability of B-FeNi-DASC. Accordingly, we can conclude that the electrocatalytic abilities of $CO_2RR$, $NO_3RR$ and urea synthesis are closely associated with the active site configurations. The $Ni-N_4$ and $Fe-N_4$ sites are selectively response to the $CO_2RR$ and $NO_3RR$, respectively, but possess negligible activity towards the urea synthesis, and the introduction of Fe-Ni pairs in the form of $FeNi-N_6$ configuration with coordinated catalytic ability is indeed favorable to urea synthesis reaction, as indicated in Fig. 2d. The amount of nitrogen in accumulated urea is much higher than that contained in the catalyst and the contrast experiments also prove that the nitrogen in urea is not derived from the electrocatalysts and the urea is electrochemically generated (Supplementary Fig. 22).

## Unraveling the origin of electrocatalytic activity and reaction mechanisms

The operando SR-FTIR measurements were carried out on B-FeNi-DASC to monitor the evolution of the bonding structure of electrochemically generated intermediate species[37]. As shown in Fig. 3, the infrared signals were collected within the wavenumber range from 1500 to 3750 $cm^{-1}$ under electrochemical conditions (potential range from −1.1 to −1.6 V versus RHE). It is observed that there are infrared bands situating at ~1978 $cm^{-1}$ and ~2170 $cm^{-1}$ corresponding to the stretching mode of N=O and C=O respectively[38,39], associating with the co-activation of nitrate ions and $CO_2$ on the catalyst, which is consistent with our aims for constructing cooperative operation of sub-reaction active sites towards $NO_3RR$ and $CO_2RR$. Meanwhile, the infrared band at 2925 $cm^{-1}$ can be assigned to stretching mode of N–H bonds[40], suggesting the moderate adsorption of *NO on B-FeNi-DASC allowing the further PCET process. It should be noted that the obvious infrared bands located at around 1694 $cm^{-1}$ is attributed to the *NHCO species and its vibration intensity increases along with applied negative potentials and reaches the maximum values at around −1.5 V, which is in good accordance with the electrochemical test results, implying that the formation intermediate species, especially *NHCO, are closely related to the urea generation[41]. On this basis, we hypothesize that the coupling of *NH with *CO plays a key role in the formation of first C–N bond of *NHCO and conducive to the urea production.

Density functional theory (DFT) calculations were performed to identify the origin of activity improvement after introducing a second metal into the single-atom catalytic system. Figure 4a lists the imposed reaction mechanism of electrochemical conversion of nitrate ions and $CO_2$ into urea on the catalyst surfaces with $NO_3^-$ as a starting point, and the reaction paths of nitrate reduction to *NO on $FeNi-N_6$ and $FeN_4-NiN_4$ are also summarized in Supplementary Fig. 24. Three pathways named NO−CO path, NOH−CO path, and NHO−CO path were considered in this work to have a systematic view of preferred urea formation routine. For single atomic Fe/Ni-doped model, the active

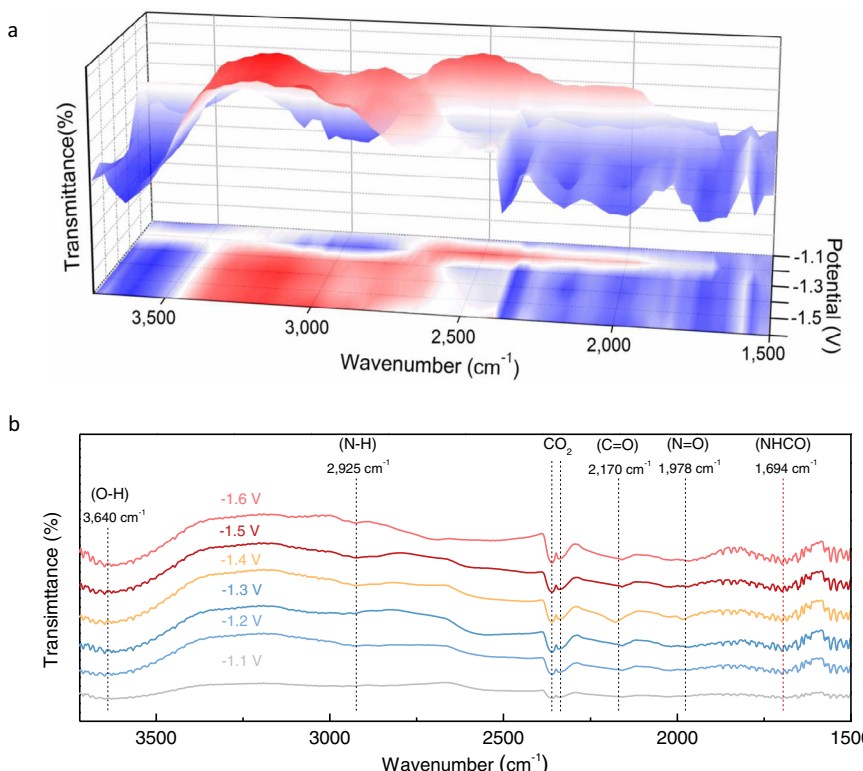

**Fig. 3 | Operando electrochemical spectroscopy measurements. a** Three-dimensional operando SR-FTIR spectra in the range of 1500–3500 cm⁻¹. **b** Infrared signal in the range of 1500–3750 cm⁻¹ under various potentials for B-FeNi-DASC during the electrocoupling of nitrate and $CO_2$.

centers were occupied by *NO intermediates, suppressing the effective capture and activation of $CO_2$. The formation free energy of key intermediate *COOH at ligand N atoms reaches up to +1.88 and +2.31 eV for Fe-N₄ and Ni-N₄ respectively (Supplementary Fig. 26), which hinders the following C–N coupling on single-atom catalysts. When introducing a second transition metal atom into system, the applied potential needed to drive the CO formation was reduced to +0.32 V, indicating the abundant active centers are of great importance in ensuring a smooth urea formation. However, the distance between two isolated transition metal sites (TM-N₄) gives the high energy barrier (+1.98 eV) in the transference of molecular CO to participate in subsequent coupling reaction (Supplementary Fig. 27), prohibiting an effective C–N coupling under ambient conditions.

The specific configuration of FeNi-N₆ as consistent in B-FeNi-DASC was constructed to evaluate its urea formation performance. The NO can be fixed on the bridge sites of Fe-Ni atoms with the adsorption energy calculated to be −2.58 eV. Although the energy barrier of direct coupling of *CO and *NO was lowered compared to the isolated Fe−Ni diatomic system, it is still energetically unfavorable (+0.81 eV), as illustrated in Supplementary Fig. 28. Noted that the chemical adsorbed *NO could be reduced to either *NOH or *HNO, the *HNO formation was more thermodynamically favorable (+0.62 eV) than that of *NOH (+0.90 eV) according to our DFT calculation results. Then the *NOH would further exothermically dissociate into *NH by accepting two electron-coupled-protons in the solvent. The limiting potential of $CO_2$ reduction on neighboring Fe atom was calculated to be −0.49 V, indicating a favorable formation of *CO. A thermodynamic spontaneous and kinetic feasible coupling between *NH and adjacent *CO with releasing of the Ni center can be observed on bonded Fe-Ni sites, and the corresponding energy barrier was calculated to be +0.21 eV (Fig. 4b), which much lower than a +1.21 eV energy barrier obtained on FeN₄−NiN₄ counterpart (Supplementary Fig. 29), as supported by the operando SR-FTIR measurements.

The second NO molecule would then be attached to the newly generated Ni site. Subsequently, the *NHCO and *NO rapidly bounded together and converted to the key intermediate of *NHCONO with an ultralow energy barrier (+0.09 eV) to overcome (Fig. 4c). The consecutive PCET processes following the formation of *NHCONO would occur to realize the urea formation finally. According to the free energy diagram, the hydrogenation of *NHCONO is the most energy-demanding step with a free energy change calculated to be +0.41 eV, from which we can conclude that the urea formation can smoothly pave after the synergistic effect of Fe−Ni diatomic pairs introduced.

The partial density of states (PDOS) of 3d orbitals of I-FeNi-DASC and B-FeNi-DASC in Supplementary Fig. 30 reveal the obvious electronic interaction near the Fermi level over FeNi-N₆ configuration. The bonded diatomic structure derived electron localization around the active site conduces to the urea generation[42–44]. It can be seen intuitively that there is a significant charge transfer in FeNi-N₆ as illustrated in the differential charge density maps. The electrons are mainly concentrated over Ni sites and the electron-deficient Fe atoms is served as the Lewis acidic sites to enhance the adsorption and activation of *NO. Nevertheless, the concentration of electrons appears both over the Fe-N₄ and Ni-N₄ sites in I-FeNi-DASC, implying the structure of Fe−Ni pairs optimizes the adsorption ability of intermediate species and C−N coupling process. The structural modeling for urea synthesis is summarized in Supplementary Figs. 30 and 31. Such moderate limiting potential and coupling barrier indicating that urea formation is thermodynamically and kinetically preferred on the B-FeNi-DASC, consistent with its excellent experimental performances.

The electrocatalytic coupling of nitrate ions with carbon dioxide for direct urea synthesis shows gigantic potential as the alternative to the traditional process. The diatomic system was proved to possess advantages towards urea synthesis in this work, owing to the synergistic catalysis with coordinated adsorption and activation of multiple reactants, particularly, the bonded Fe−Ni configurations serve as the

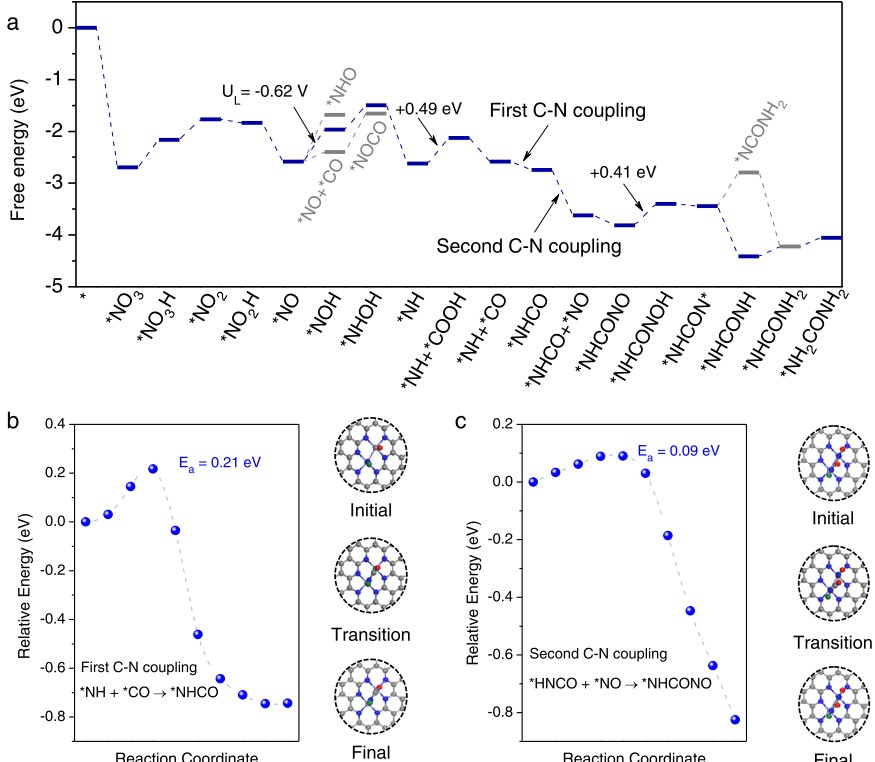

**Fig. 4 | Thermodynamic and kinetic calculation results for urea synthesis.** **a** Free energy diagram of urea production. **b** The reaction pathway of first C–N coupling for *NHCO formation and **c** second C–N coupling for *NHCONO formation. The structures of the initial, transition and final states along with the *NHCO and *NHCONO formation are also presented. The purple, indigo, blue, red, and gray balls represent Fe, Ni, N, O, and C atoms, respectively.

efficient sites for C–N coupling, to give overall optimization from the generation and coupling of intermediate species. The two steps of chemical coupling to generate C–N bonds were demonstrated to be thermodynamic spontaneous and kinetic favorable over Fe–Ni pairs. The current work demonstrated that the efficient urea synthesis on a bonded Fe–Ni pairs enriched diatomic electrocatalyst with a high urea yield rate of 20.2 mmol h$^{-1}$ g$^{-1}$ and FE of 17.8%. The competing hydrogen evolution was in situ suppressed with a total FE of about 100% for the value-added product formation of CO, NH$_3$, and urea. This work focuses on the identifying and tailoring the C–N coupling site at atomic level, expected to guide the further development direction towards efficient electrocatalytic urea synthesis.

# Methods
## Materials fabrication
The synthesis of single atom and diatomic electrocatalysts is described elsewhere[45]. N-coordinated transition metal atomic sites anchoring on porous carbon framework were fabricated via pyrolysis of coordination polymer. Typically, 0.37 g 2,4-diaminopyrimidine was dissolved into 50 ml ethanol, followed by the addition of 0.50 g 2,6-diacetylpyridine and 1 ml acetic acid. Subsequently, the mixed clear solution was heated up to 60 °C for 12 h to obtain polymeric bis(imino)pyridine precipitates. Then, 1.0 g FeCl$_2$ dissolved in 10 ml ethanol was added into the above dispersion and stirred for another 12 h. The obtained dried powder was carbonized under flowing N$_2$ for 1 h at 900 °C. The product underwent acid leaching (0.5 M H$_2$SO$_4$) successively remove unstable metallic species. Finally, the Fe-SAC was obtained by second thermal treatment at 900 °C for another 1 h. The synthetic procedures of Ni-SAC are similar to that of Fe-SAC except that 0.79 g NiCl$_2$ was used, and the I-FeNi-DASC was obtained with the addition of 0.50 g FeCl$_2$ and 0.40 g NiCl$_2$ simultaneously. The diatomic electrocatalyst with bonded Fe–Ni pairs (B-FeNi-DASC) was synthesized via a host-guest strategy. The Fe$^{2+}$-coordinated polymer was obtained by reaction

of bis(imino)pyridine precipitate with 0.50 g FeCl$_2$ initially. The obtained powder was dried and then redispersed in n-hexane, followed by the addition of metal ions solution containing 0.40 g NiCl$_2$. The post-adsorbed Ni ions would form bonding with the neighboring Fe nodes and the subsequent thermal treatment induces the formation of bonded Fe–Ni site configuration. The M-FeNi-DASC was fabricated by physically mixing of Fe-SAC and Ni-SAC during the preparation of the catalyst slurry.

## Materials characterization
The morphologies of electrocatalysts were characterized by scanning electron microscopy (SEM and FESEM) using a SU8220 field emission scanning electron microscope and TEM using Titan ETEM G2 80-300 electron microscope operating at 300 kV with electron energy loss microspectroscopy (Gatan model 965). The XAS were recorded at the XAFCA beam of the Singapore Synchrotron Light Source in transmission mode. The XAS on Fe, Ni K-edge of the products in various synthetic steps were recorded at XAFCA beamline of Singapore Synchrotron Light Source using a Vertox ME4 silicon drift diode detector in fluorescence mode with Au as the reference for calibration. X-ray diffraction (XRD) was carried out on a Smart Lab diffractometer using a Cu Kα (λ = 1.5405 Å) radiation source (Rigaku Co.). Raman spectra were measured on a Lab Ram HR Evolution Raman spectrometer with a 532 nm wavelength incident laser light (Horiba Jobin Yvon Co.). Brunauer-Emmett-Teller (BET) surface area and pore size distribution of catalysts was analyzed using a Micromeritics instrument (CORP ASAP 2460). X-ray photoelectron spectroscopy (XPS) measurements were carried out on an ESCALAB 250 system to determine the composition and chemical states of the catalysts. Operando SR-FTIR measurements were made at the infrared beamline BL01B of the National Synchrotron Radiation Laboratory (NSRL, China) through a homemade top-plate cell reflection infrared set-up with a ZnSe crystal as the infrared

transmission window (cut-off energy of ~625 cm$^{-1}$).This end station was equipped with an FTIR spectrometer (Bruker 66 v s$^{-1}$) with a KBr beam splitter and various detectors (herein a liquid nitrogen cooled mercury cadmium telluride detector was used) coupled with an infrared microscope (Bruker Hyperion 3000) with a ×16 objective, and could provide a high spectral resolution of 0.25 cm$^{-1}$.

## Electrochemical measurements

The electrochemical test in the H cell was carried out on a CHI 660E electrochemical station in a three-electrode configuration. The pretreated Nafion 117 membrane (Dupont) served as the separator, and the electrolyte used in this work was 0.1 M KHCO$_3$ with 50 Mm KNO$_3$ or KNO$_2$. KHCO$_3$ (≥99.99% metals basis, 99.7–100.5% dry basis) was bought from Aladdin Biochemical Technology Co., Ltd. KNO$_3$ (>99%) were received from Sinopharm Chemical Reagent Co., Ltd. Catalyst (2 mg) was dispersed in 950 μl of Isopropyl alcohol and 50 μl of Nafion (5 wt% aqueous solution) with sonication for 30 min to form a homogenous ink. Then, 100 μl of catalyst ink was loaded onto a piece of carbon paper (Hesen) and dried naturally to obtain the working electrode; the geometric area of the working electrode was 1 × 1 cm$^2$, carbon papers (HCP020N) were purchased from Hesen Company (Shanghai, China) and washed with water and acetone before use. And the catalyst loading was 0.2 mg cm$^{-2}$. The reference electrode was an Ag/AgCl electrode containing saturated KCl solution, and a carbon rod served as the counter electrode. Before electrochemical tests, the cathode part of the electrolyte was purged with the corresponding gases for pre-saturation. Ar (>99.999%), N$_2$ (>99.999%), CO$_2$ (>99.999%) were bought from Changsha Rizhen Gas Co., Ltd. All chemicals were used without further purification. After that, the flow rate was maintained at 30 mL min$^{-1}$ during the catalytic process. The provided applied potentials were against an Ag/AgCl reference electrode (saturated KCl solution) and converted to the RHE reference scale using $E_{RHE} = E_{Ag/AgCl} + 0.0591 \times pH + 0.197$. The pH value is 6.8 or 8.3 for the measurement with or without the CO$_2$ saturation, respectively. The scan rate for linear sweep voltammetry (LSV) tests is 10 mV s$^{-1}$.

## Product quantification

The gaseous products were analyzed online with a SP-7820 Gas chromatography. The product of ammonia was quantified by the indophenol blue method with the coloring agents of a, 1 M NaOH solution containing 5 wt% sodium citrate and 5 wt% salicylic acid; b, 0.05 M NaClO solution; c, 1 wt% sodium nitroferricyanide solution. 2 mL of electrolyte was extracted and 2 mL of a, 1 mL of b and 0.2 mL of c was added into the electrolyte in turn, then the mixture was kept under dark for 2 h before measurement of the absorbance[46]. The absorbance at 660 nm exhibits the linear relationship with the concentration of ammonia, thus the amount of ammonia can be obtained based on the calibration curve. The concentration of urea was quantified by the urease method[47,48]. Urease from Canavalia ensiformis (Jack bean) was purchased from Sigma, Urease activity: 20KU, Batch number: Lot#SLCJ5647.

The decomposition of urea by urease (Sigma, 5 mg mL$^{-1}$) was conducted at 40 °C for 0.5 h, the mole weights of ammonia before and after the decomposition experiment were quantified by the above method and were denoted as $m_b$ and $m_a$, respectively. Since 1 mole of urea can be decomposed into 1 mole of CO$_2$ and 2 moles of NH$_3$, the mole weight of yielded urea ($m_{urea}$) can be calculated as followed:

$$m_{urea} = \frac{m_a - m_b}{2} \qquad (1)$$

The Faradaic efficiency for electrocatalytic urea synthesis was obtained by the following equation:

$$FE(\%) = \frac{n \times F \times C \times V}{60.06 \times Q} \times 100 \qquad (2)$$

where $F$ is the Faraday constant, $Q$ is the electric quantity, $C$ is the concentration of generated urea and $V$ is the volume of the electrolyte, $n$ is the number of electron transfer in the electrochemical reaction and it's 12 for electro-coupling of CO$_2$ and nitrite and 16 for electro-coupling of CO$_2$ and nitrate. The formation rate of urea was averaged by the time, and the presented urea formation rate was the average value within the time frame of the tests.

## NMR measurements

The conditions for $^{15}$N-labeling electrochemical measurements are the same with the previous tests except that the $^{14}$KNO$_3$ was substituted by $^{15}$KNO$_3$ (Isotopic abundance: 99 atom%, Batch number: Lot#F2123307, Aladdin Biochemical Technology Co., Ltd. Shanghai). The produced urea products were decomposed into ammonia by urease (Urease from Canavalia ensiformis, Urease activity: 20KU, Batch number: Lot#SLCJ5647, Sigma), and the pH value of the solution was adjusted to ~3 before NMR measurement. For the detection of ammonia by-product with the co-existence of urea, the tests were carried out immediately after acidification to avoid the urea decomposition. As a typical NMR test process, 500 μL of electrolyte was extracted, followed by the additions of 100 μL d$_6$-DMSO (99.9 atom% D, Innochem Technology Co., Ltd. Beijing) as the deuterated solvent and 50 μL DSS sodium salt solution (Sodium 3-(Trimethylsilyl)−1-propanesulfonate, Concentration: 5.0 mM, Batch number: Lot#M-2076, Cambridge Isotope Laboratories, Inc) as the internal standard. The presented data is the accumulated result of 256 scans on a 600 MHz NMR instrument (Bruker) equipped with an ultra-low temperature probe. The urease decomposition-NMR method affords an indirect approach for identification and quantification of urea. On the other hand, the direct NMR method was adopted to detect the urea directly on an 800 MHz NMR instrument (Bruker) equipped with an ultra-low temperature probe, and the presented data is the accumulated result of 1024 scans. Typically, 500 μL of urea-containing electrolyte was mixed with 100 μL d$_6$-DMSO and 50 μL of aforementioned DSS sodium salt solution, and then the NMR measurements were conducted without post-treatment.

## Computational method

The DFT computations were performed via the Vienna ab initio simulation package (VASP)[49]. The ion-electron interactions were described with the projector-augmented plane-wave (PAW) method[50]. Exchange-correlation potentials were expressed by Perdew-Burke-Ernzerhof (PBE) functional with the generalized gradient approximation[51]. The dual-metal FeNi−N$_6$−C was constructed on a 5 × 6 graphene supercell, cutoff energy for geometry optimization was set to be 460 eV, and the Brillouin zone was sampled with 3 × 3 × 1. To avoid the interlayer interaction the vacuum layer perpendicular to the FeNi-N$_6$ and FeN$_4$−Ni−N$_4$ slabs was set to be 20 Å. The systems were relaxed until the energy and force reaching the convergence threshold of 10$^{-5}$ eV and 0.01 eV Å$^{-1}$. The solvent effect was taken into consideration by imposing an implicit solvent model with the electric constant set to be 80[52]. The climbing-image nudged elastic band (CI-NEB) method[53] was used to obtain the energy barrier of the non-electrochemical electron coupling process. The snapshot of optimized geometry structures of initial state (IS), transition state (TS) and finial state (FS) along the reaction paths were listed in Supplementary Information.

## Data availability

All data generated or analyzed during this study are included in this Article (and Supplementary Information). Data for Figs. 1–4 are available as source data with this paper. Source data are provided with this paper.

## Code availability

The computational codes used in the current work are available from the corresponding author on reasonable request.

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

## Acknowledgements

The authors acknowledge the Singapore Synchrotron Light Source, National Synchrotron Radiation Laboratory for technical support and the Nanjing Normal University for supporting the computational work. Wang acknowledges financial support from the National Key R&D Program of China (2020YFA0710000), Jiang acknowledges financial support from the National Natural Science Foundation of China (Grant No. 21573066, 21902047) and Australia Research Council (DP180100731, DP180100568). Chen acknowledges financial support from the China Postdoctoral Science Foundation (Grant No. BX20200116 and 2020M682540) and the National Natural Science Foundation of China (22102054). Zheng acknowledges financial support from the National Natural Science Foundation of China (22075075) and the Provincial Natural Science Foundation of Hunan (2021RC3051).

## Author contributions

S.W., C.C., S.P.J., and J.Z. conceived the project. X.Z. and C.C. carried out most of the materials fabrication and characterization and prepared the manuscript. X.Z. and Y.L. performed the theoretical calculations. S.B. and Q.L. conducted the operando SR-FTIR measurements. M.Q., X.W., and N.H. conducted part of the electrochemical measurements. C.X., W.C., and P.C. performed part of the physical characterization of the catalysts. All authors discussed the results and commented on the manuscript.

## Competing interests

The authors declare no competing interests.
