## [Peer review file · Nature Communications]

REVIEWER COMMENTS

Reviewer #1 (Remarks to the Author):

Through this article entitled "Identifying and tailoring C-N coupling sites for efficient urea synthesis on diatomic Fe-Ni catalyst" the authors have attempted to provide a solution to a very challenging problem of direct synthesis of urea through coupling of Carbon and Nitrogen (bypassing the harsh reaction conditions of the Haber-Bosch process of synthesis of NH₃). They have synthesized the catalytic systems and studied the urea synthesis paths. Their work is also validated by DFT calculations. But there are some ambiguities and addressing those in the revised manuscript will surely increase the impact of this manuscript.

1. They have concluded "The current work demonstrated for the first time the efficient urea synthesis on a bonded Fe-Ni pairs enriched diatomic electrocatalyst with a high urea yield rate of 20.2 mmol h⁻¹ g⁻¹ and FE of 17.8%. The competing hydrogen evolution was in-situ suppressed with a total FE of about 100% for the value-added product formation of CO, NH₃ and urea."

In the second sentence they have indicated total 100% FE for CO, NH₃ and urea formation over HER. What is the total Faradaic efficiency? Please clarify.

2. They have indicated that over the bonded Fe-Ni-DASC the FE of CO, NH₃ and urea formation are 73.3%, 77.6% and 17.8% respectively at -1.5V. Will CO and NH₃ formation not compete with urea formation? If the reactions compete how can this study provide the efficient route for urea synthesis suppressing those reactions?

3. It is observed that urea synthesis is performed in presence of nitrate or nitrite and CO₂. But in SR-FTIR measurements there is a sign of moderate NO adsorption on the B-FeNi-DASC. They have also theoretically studied urea synthesis from NO and CO₂ over the same catalytic surface. What is the source of this adsorbed NO? If it is obtained from NO₃⁻ or NO₂⁻ please include the relative free energy diagram of NO formation.

4. Please indicate the theoretical adsorption energy and/or free energy of adsorption of CO₂, NO, nitrate, nitrite and CO over the SACs and isolated/bonded-Fe-Ni-DASCs.

5. Please mention the relative free energies of all the adsorbed species and the intermediates in the free energy diagram in Figure-4a.

6. Please include the clear side and above views of the electronic structures of the adsorbed species and intermediates over the catalytic surface, so that a clear overview of the structures can be perceived.

7. In Supplementary Figure-23 authors have presented the differential charge density maps of the isolated/bonded-DASCs. Please indicate the iso-value of these maps.

8. In this context the following articles can be consulted for discussion,

a. <https://dx.doi.org/10.1021/acs.jpcclett.1c00421>

b. <https://dx.doi.org/10.1021/acs.jpcclett.1c03242>

c. <https://dx.doi.org/10.1021/acs.jpcc.1c08779>

Reviewer #2 (Remarks to the Author):

In this manuscript, the authors demonstrated an important strategy of oxygen vacancies towards electrocatalytic urea synthesis, and clarified the reaction mechanism. It was found that oxygen vacancies inhibited the hydrogenation reaction by anchoring the *NO intermediates, thereby promoting the C-N coupling process. The reaction mechanism was supported by in situ electrochemical characterization via tracing the reaction intermediate species. Since the precise quantification of urea is an effective prerequisite for the development of this field, the authors have investigated the potential impact of multiple by-products on the urea quantification, and demonstrated the most reliable urea quantification protocol, contributing to the development of related fields. Overall, this work is interesting with insightful information. The reviewer would like to recommend its acceptance after addressing the following technical points.

1. The authors demonstrated that the urea decomposition method is the most suitable one to quantify the electrocatalytic generated urea. Under certain urea concentration, whether the ammonia obtained by urease decomposition is reliable or not? The author needs to provide the source of urease, including the manufacturer and production batch, so that other researchers could reproduce the results more easily.

2. The authors utilized the in-situ SFG to detect the intermediate during the electrocatalytic process. The optical image of equipment and the schematic diagram of working principle could be provided in the supporting information as the individual Supporting Figure, which may help others to reproduce the experiments.

3. The details for the quantification of nitrite ions in the electrolytes should be supplemented.

4. The authors have evaluated the accuracy of the quantitative methods of the products, and concluded that the diacetyl monoxime method was susceptible to interference of nitrite ions and the urease decomposition method was the reliable one. However, the influence of carbon-containing products on the detection method was not considered, for example, common formic acid, methanol and ethanol. Further evaluation of the impact of the above possible products on quantitative methods is needed to expand the scope of application of the conclusion.

5. In the Supporting Information, “In site sum frequency generation” should be “In situ sum frequency generation”. In the Figure S5d and the Figure S6, the units for the abscissa are missing, which needed to be supplemented. The authors should also check the whole manuscript to correct some of typos.

Reviewer #3 (Remarks to the Author):

See attached

Reviewer #4 (Remarks to the Author):

The electrocatalytic C-N coupling serves as a promising alternative to the traditional urea synthesis and the emerging field of electrocatalysis research. In this work, the authors designed a diatomic catalyst with bonded Fe-Ni pairs to improve the efficiency of electrochemical urea synthesis. The “three-in-one” of active site, activation site and coupling site was realized in the FeNi-N6 sites. The origin of the superior electrocatalytic activity for FeNi-N6 was experimentally and theoretically confirmed. This innovative research work achieved the identification and tailoring of C-N coupling sites for electrocatalytic urea synthesis, which is expected to promote the green revolution of urea industrial and provide broaden guidance for catalytic coupling reactions. Considering the significance of this work in the related fields, I would like to suggest its publication in Nature Communications after a minor revision.

1. The information of the materials used must be afforded in the section of methods, including the purity of carbon dioxide gas, the manufacturer and purity of chemical reagents.

2. The urease decomposition method was adopted in this work for the quantification of urea product. The UV-vis adsorption curves for the urea quantification need to be complemented in the Supplementary Information.
3. The structural stability is highly important for the recognition of active sites and the evaluation of catalyst stability. The author needs to confirm whether there is metal leaching under the working condition.
4. The quality of Figures needs to be further improved, such as Figure 1c and Figure 1d.
5. There are some errors displayed in Figure 2b. The authors should check this point and the expressions in the manuscript.

Review comments:

Communications manuscript NCOMMS-22-16001

General comments for XANES:

Except edge position drifting, other XANES features have been experimentally revealed by the reported XANES data, which is yet not fully addressed. Fe and Ni K edges XANES reported by SI Fig.6 (a) & (b) reveal well resolved XANES difference throughout the sample system in (1) pre-edge feature, (2) lineshape throughout the edge jump, (3) peak drifting, peak splitting, overall lineshape changing, and peak amplitude varying of XANES whiteline, and (4) significant difference in the post whiteline XANES region. These well resolved features are specific to the Fe and Ni local coordination environments, featured by their occupied sites, respectively. It is suggested to further address these experimentally revealed XANES features, correlating them to the local structural environment for the Fe and Ni occupied site predicted by DFT.

Some other further comments:

Table 1.

	Related contents	comments
Manuscript, Page 3	peak position situates at lower energy than those of Fe foil...	A typo: "lower" should be replaced by "higher".
	X-ray absorption fine structure (FT-EXAFS) analysis	There is no analysis here, simply presenting magnitude of FT.
	peak at approximate 2.5 Å	Difficult to identify where is peak for B-FeNi-DASC, suggesting to arrow indicated.
	proving the successful fabrication of isolated and bonded diatomic site configurations	EXAFS can only provide 1D info in terms of R and CN. Not info about the band angle is completely missing. Thus, it cannot fully prove the "site configurations"
SI, page 8	Supplementary Figure 7	SI Fig. 7(a) vs (b): Better quality in EXAFS R space curve fitting was obtained upon the 2 nd shell between ~R2-3Å for Ni K edge data vs that for Fe K edge regarding sample B-FeNi-DASC. Please address whether the difference in fitting quality between two fitting is induced by the defect of the DFT model, which was used to guide the reported fitting, either structurally and/or chemically, or induced by other reason(s)? SI Fig.7(d): Data and Feff fit match reasonably well for the imaginary part of FT of Ni K edge EXAFS for the outer shell region ~R2.5-4.0Å, but there is well-resolved difference for the magnitude of FT between Data and Feff fit throughout this outer R region.

		Please address what is the reason for this inconsistency in fitting quality between the magnitude of FT and the imaginary part of FT? What is the impact of the revealed fitting defect to those fitted structure parameters for the corresponding outer shell Ni coordination (SI table 1)? The accuracy for which fitted parameter(s) is impacted, and in what scale? Fig. 7(e): certain fitting defect is revealed for Fe outer shell fit for the imaginary part of FT for region $\sim R(2.5-3.0\text{\AA})$. Please clarify whether this is a structure effect or chemistry effect?
SI page 2	Supplementary Table 1	 1. Error bars are missing for all reported parameters; 2. If coordination numbers (N) are all fixed throughout fitting for all scattering paths, either fixed upon crystallography of metallic Fe/Ni, and/or to what DFT predicted, please specify in the table.

Point-by-point responses and revisions made in the manuscript (NCOMMS-22-16001)

Reviewer #1 (Remarks to the Author):

Through this article entitled “Identifying and tailoring C-N coupling sites for efficient urea synthesis on diatomic Fe-Ni catalyst” the authors have attempted to provide a solution to a very challenging problem of direct synthesis of urea through coupling of Carbon and Nitrogen (bypassing the harsh reaction conditions of the Haber-Bosch process of synthesis of NH_3). They have synthesized the catalytic systems and studied the urea synthesis paths. Their work is also validated by DFT calculations. But there are some ambiguities and addressing those in the revised manuscript will surely increase the impact of this manuscript.

Thanks for your approval to this paper and we have made corresponding revisions according to your instructive comments, the details are afforded as followed.

1. They have concluded “The current work demonstrated for the first time the efficient urea synthesis on a bonded Fe-Ni pairs enriched diatomic electrocatalyst with a high urea yield rate of $20.2 \text{ mmol h}^{-1} \text{ g}^{-1}$ and FE of 17.8%. The competing hydrogen evolution was in-situ suppressed with a total FE of about 100% for the value-added product formation of CO , NH_3 and urea.” In the second sentence they have indicated total 100% FE for CO , NH_3 and urea formation over HER. What is the total Faradaic efficiency? Please clarify.

Response:

We are really appreciating for your instructive comments. The above expressions regarding “total Faradaic efficiency” may cause confusion for the readers. The total Faradaic efficiency in this work refers to the efficiency for the formation of value-added CO , NH_3 and urea products.

In the revised manuscript, the sentence of “The competing hydrogen evolution is *in-situ* suppressed by C-N coupling with a total Faradaic efficiency of about 100% for the value-added products.” has been revised to “A total Faradaic efficiency of about 100% for the formation of value-added urea, CO and NH_3 was realized.”

The sentence of “Meanwhile, the total FE of the formation of the urea, CO and NH₃ reaches about 100%, implying that the competing HER over B-FeNi-DASC system is successfully *in-situ* suppressed during the coupling reaction.” has been corrected to “Meanwhile, the total FE for the formation of value-added CO, NH₃ and urea products reaches about 100%, negligible hydrogen was produced in the co-electrolysis measurement over B-FeNi-DASC system, implying that the competing HER is almost completely suppressed.”

2. They have indicated that over the bonded Fe-Ni-DASC the FE of CO, NH₃ and urea formation are 73.3%, 77.6% and 17.8% respectively at -1.5 V. Will CO and NH₃ formation not compete with urea formation? If the reactions compete how this study can provide the efficient route for urea synthesis suppressing those reactions?

Response:

Thanks for your comments. The electrocatalytic urea synthesis involves the co-activation of carbon dioxide and nitrate. High carbon dioxide or nitrate reduction activity alone leads to selective activation of reactive species on the catalyst surface, as a result, the reaction rate and the concentration of active intermediate species (carbon-containing and nitrogen-containing) involved in the coupling reaction do not match, resulting in the low urea synthesis activity for the single-atom electrocatalysts. Therefore, we designed a diatomic catalyst (FeNi-N₆) with good catalytic activity for both carbon dioxide reduction and nitrate reduction. During co-electrolysis measurements, the considerable amount of *CO and *NH species can be formed on the surface. More importantly, the thermodynamic spontaneous reaction between *CO and *NH conducive to the C-N coupling and subsequent urea synthesis.

The formation of CO and NH₃ provides direct evidence for the generation of carbon-containing and nitrogen-containing intermediate species, and the intermediate species were utilized and coupled to synthesize urea through the thermodynamic and kinetic feasible C-N coupling reactions. The subsequent researches can be aimed at the design and optimization in terms of the geometric density of coupling sites and the migration efficiency of intermediate species.

3. It is observed that urea synthesis is performed in presence of nitrate or nitrite and CO₂. But in SR-FTIR measurements there is a sign of moderate NO adsorption on the B-FeNi-DASC. They have also theoretically studied urea synthesis from NO and CO₂ over the same catalytic surface. What is the source of this adsorbed NO? If it is obtained from NO₃⁻ or NO₂⁻ please include the relative free energy diagram of NO formation.

Response:

We thank the reviewer for raising such a meaningful question. The source of this adsorbed NO is nitrate reduction reaction. In our revised manuscript, we calculated the free energy diagram of NO formation with NO₃⁻ as a starting point.^[R2,R3] Infer from the Supplementary Figure 24, we find out that the potential limiting step for NO formation on FeNi-N₆ and FeN₄-NiN₄ is the first hydrogenation process. (*NO₃ + H⁺ + e⁻ → *NO₃H), corresponding U_L are -0.53 V and -2.29 V for FeNi-N₆ and FeN₄-NiN₄ respectively. The weak *HNO₃ adsorption on FeN₄-NiN₄ are too weak to drive the NO formation while the synergy effect between Fe-Ni atoms in FeNi-N₆ greatly enhanced the adsorption strength of *HNO₃, making the direct formation of NO feasible.

The following words “Infer from the Supplementary Figure 24, we find out that the potential limiting step for NO formation on FeNi-N₆ and FeN₄-NiN₄ is the first hydrogenation process. (*NO₃+H⁺+e⁻→*NO₃H), corresponding U_L are -0.53 V and -2.29 V for FeNi-N₆ and FeN₄-NiN₄ respectively. The weak *HNO₃ adsorption on FeN₄-NiN₄ are too weak to drive the NO formation while the synergy effect between Fe-Ni atoms in FeNi-N₆ greatly enhanced the adsorption strength of *HNO₃, making the direct formation of NO feasible.” have been added into the revised Supplementary Information.

The following figure have been supplemented as the Supplementary Figure 24.

[R2] Chen, G. et al. Electrochemical reduction of nitrate to ammonia via direct eight-electron transfer using a copper-molecular solid catalyst. *Nat. Energy* **5**, 605-613 (2020).

[R3] Wang, J. et al. Electrocatalytic reduction of nitrate to ammonia on low-cost ultrathin CoO_x nanosheets. *ACS Catal.* **11**, 15135-15140 (2021).

The reaction path of NO₃⁻ reduction to *NO follows the above references.

Supplementary Figure 24 | The free energy diagram of NO formation with NO₃⁻ as the starting point on FeNi-N₆ and FeN₄-NiN₄ slabs.

4. Please indicate the theoretical adsorption energy and/or free energy of adsorption of CO₂, NO, nitrate, nitrite and CO over the SACs and isolated/bonded-Fe-Ni-DASCs.

Response:

We thank the reviewer for pointing this out. In our revised manuscript, the following words “The adsorption energies of CO₂, NO, nitrate, nitrite, and CO over the SACs and isolated/bonded-Fe-Ni-DASCs have been listed in Supplementary Figure 25, supporting the superior urea synthesis performance of FeNi-N₆ pair site.” have been added in the revised Supplementary Information.

The following figure has been added in the revised version as **Supplementary Figure 25**.

Supplementary Figure 25 | The adsorption energy of nitrate, nitrite, NO, CO₂ and CO on single-atom electrocatalysts (FeN₄, NiN₄) and diatomic electrocatalysts (FeNi-N₆, FeN₄-NiN₄) respectively.

5. Please mention the relative free energies of all the adsorbed species and the intermediates in the free energy diagram in Figure 4a.

Response:

Thank you for reminding us the discipline of data presentation. The free energy diagram of nitrate reduction to *NO has been supplemented. The relative free energies of all intermediates involved along the urea formation are clearly checked and added to the revised Figure 4a.

Fig. 4 | Thermodynamic and kinetic calculation results for urea synthesis. **a**, Free energy diagram of urea production.

6. Please include the clear side and above views of the electronic structures of the adsorbed species and intermediates over the catalytic surface, so that a clear overview of the structures can be perceived.

Response:

Thank you for your comments. We added the clear side and top view of structures of intermediates involved urea formation over the FeNi-N₆ surface with their name marked in our revised manuscript. The red, gray, blue, purple, orange and cyan balls represent O, C, N, Fe, H and Ni atoms respectively.

The following figure has been supplemented in the revised version as Supplementary Figure 32.

Supplementary Figure 32 | The top and side view of intermediates involved in urea formation on FeNi-N₆ slab. The red, gray, blue, purple, orange and cyan balls represent O, C, N, Fe, H and Ni atoms respectively.

7. In Supplementary Figure 31, authors have presented the differential charge density maps of the isolated/bonded-DASCs. Please indicate the iso-value of these maps.

Response:

Thank you for raising this important issue. We supplied the iso-value of the differential charge density maps in our revised manuscript.

The words “The charge density difference with iso surface value of 0.01 e/Bohr³ of FeNi-N₆ and FeN₄-NiN₄ respectively. Blue and yellow colors represent losing and gaining electrons respectively.” have been supplemented in Supplementary Figure 30.

8. In this context the following articles can be consulted for discussion, a.<https://dx.doi.org/10.1021/acs.jpcllett.1c00421>. b.<https://dx.doi.org/10.1021/acs.jpcllett.1c03242>. c.<https://dx.doi.org/10.1021/acs.jpcc.1c08779>.

Response:

Many thanks to the reviewers for providing such crucial literature information, which is very helpful for us to deepen the reaction mechanism of electrocatalytic urea synthesis, therefore, we have cited these published related articles as Ref. 42, Ref. 43 and Ref. 44 in the theoretical discussion part of revised version.

42. Roy, P., Pramanik, A. & Sarkar, P. Dual-silicon-doped graphitic carbon nitride sheet: An efficient metal-free electrocatalyst for urea synthesis. *J. Phys. Chem. Lett.* 2021, **12**, 10837-10844 (2021).

43. Roy, P., Pramanik, A. & Sarkar, P. Graphitic carbon nitride sheet supported single-atom metal-free photocatalyst for oxygen reduction reaction: A first-principles analysis. *J. Phys. Chem. Lett.* **12**, 2788-2795 (2021).

44. Ball, B., Das, P. & Sarkar, P. Molybdenum atom-mediated salphen-based covalent organic framework as a promising electrocatalyst for the nitrogen reduction reaction: A first-principles study. *J. Phys. Chem. C* 2021, **125**, 26061-26072 (2021).

Reviewer #2 (Remarks to the Author):

In this manuscript, the authors demonstrated an important strategy of oxygen vacancies towards electrocatalytic urea synthesis, and clarified the reaction mechanism. It was found that oxygen vacancies inhibited the hydrogenation reaction by anchoring the *NO intermediates, thereby promoting the C-N coupling process. The reaction mechanism was supported by in situ electrochemical characterization via tracing the reaction intermediate species. Since the precise quantification of urea is an effective prerequisite for the development of this field, the authors have investigated the potential impact of multiple by-products on the urea quantification, and demonstrated the most reliable urea quantification protocol, contributing to the development of related fields. Overall, this work is interesting with insightful information. The reviewer would like to recommend its acceptance after addressing the following technical points.

Much appreciate for your approval to our work. We have supplemented the data and explanations to address the mentioned technical points.

1. The authors demonstrated that the urea decomposition method is the most suitable one to quantify the electrocatalytic generated urea. Under certain urea concentration, whether the ammonia obtained by urease decomposition is reliable or not? The author needs to provide the source of urease, including the manufacturer and production batch, so that other researchers could reproduce the results more easily.

Response:

Thanks for your comments. The urease activation and dosage are applied for fully decomposition of urea into ammonia, and the subsequent ammonia quantification is not be disturbed in this work. In order to verify the reliability of urease decomposition method used in this work. We prepared a series of urea standard solutions (concentration of 5 ppm, close to experimental value), these urea standard solutions (10 batches) were decomposed into ammonia by urease with the same conditions. The concentration of ammonia obtained by the indophenol blue method is listed in Figure R1b. It was found that the ammonia concentrations from 10 batches remain unchanged, indicating that the urea-derived ammonia quantification method with urease decomposition is stable. The urea

quantification results by urease decomposition-indophenol blue method are matched with that of results by urease decomposition-NMR method, as illustrated in Supplementary Figure 17, implying the applicability of urease decomposition for urea quantification. More important, the recent report also demonstrated that the urease decomposition-indophenol method is the reliable one for quantifying urea without interference from coexisting byproducts (*J. Am. Chem. Soc.* 2022, **144**, 11530-11535).

The manufacturer and production batch of urease were list and added in revised manuscript. The following words “Chemicals. Carbon papers (HCP020N) were purchased from Hesen Company (Shanghai, China) and washed with water and acetone before use. KHCO_3 ($\geq 99.99\%$ metals basis, 99.7-100.5% dry basis) was bought from Aladdin Biochemical Technology Co., Ltd. KNO_3 ($>99\%$) were received from Sinopharm Chemical Reagent Co., Ltd. Urease from *Canavalia ensiformis* (Jack bean) was purchased from Sigma, Urease activity: 20KU, Batch number: Lot#SLCJ5647. Ar ($>99.999\%$), N_2 ($>99.999\%$), CO_2 ($>99.999\%$) were bought from Changsha Rizhen Gas Co., Ltd. All chemicals were used without further purification.” have been added in Methods section in revised manuscript.

The following literature has been cited in the revised manuscript as (Ref. 47) to clarify the adopted urea quantification method is reliable in this work.

47. Wei, X. et al. Oxygen vacancy-mediated selective C-N coupling toward electrocatalytic urea synthesis. *J. Am. Chem. Soc.* **144**, 11530-11535 (2022).

Figure R1 | (a) The corresponding absorbance curve by UV-vis spectrophotometer with and without urease decomposition of the urea standard solution. The absorbance at 662 nm was measured by UV-vis spectrophotometer. (b) The corresponding ammonia concentration of urea standard solution decomposed by urease from 10 batches.

Supplementary Figure 17 | The identification and quantification for electrochemical ammonia synthesis. (a) The ^1H NMR spectra for urea product-derived ammonia at -1.4 V versus RHE for isotope labeling measurements. (b) The urea yield rates and corresponding Faradaic efficiencies obtained by urease decomposition-indophenol blue method and urease decomposition-NMR method for isotope labeling measurements at -1.4 V versus RHE.

2. The authors utilized the in-situ SFG to detect the intermediate during the electrocatalytic process. The optical image of equipment and the schematic diagram of working principle could be provided in the supporting information as the individual Supporting Figure, which may help others to reproduce the experiments.

Response:

The following figure has been added to the Supplementary Information as Supplementary Figure 23 in the revised version.

Supplementary Figure 23 | The optical images of instrument for in-situ electrochemical spectroscopy characterization.

3. The details for the quantification of nitrite ions in the electrolytes should be supplemented.

Response:

Thanks for your comments. A series of NO_2^- with various concentrations were prepared and corresponding calibration curves were obtained in Supplementary Figure 34.

The nitrite quantification details of " NO_2^- was determined using N-(1-naphthyl)-ethylenediamine dihydrochloride spectrophotometric method with some modification. Desaiily, 0.5 g of sulfanilic acid was dissolved in the mixture of 90 mL of H_2O and 5 mL of acetic acid. Then, 5 mg of n-(1-naphthyl)-ethylenediamine dihydrochloride was added and the solution was filled to 100 mL to obtain chromogenic agent. 1 mL of the treated electrolyte was mixed with 4 mL of chromogenic agent and kept in dark for 15 min. The UV-vis absorption spectrum was then acquired at 540 nm." have been supplemented in the Supplementary Information.

Supplementary Figure 34 | (a) A series of standard solutions with NO_2^- concentrations of 0 ppm, 1.0 ppm, 2.5 ppm, 5.0 ppm and 10.0 ppm respectively. (b) Calibration curve used for quantification of NO_2^- concentration. The absorbance at 540 nm was measured by UV-vis spectrophotometer, and the fitting curve shows good linear relation of absorbance with ammonia concentration ($y = 0.3106x + 0.041$, $R^2 = 0.9998$).

4. The authors have evaluated the accuracy of the quantitative methods of the products, and concluded that the diacetyl monoxime method was susceptible to interference of nitrite ions and the urease decomposition method was the reliable one. However, the influence of carbon-containing products on the detection method was not considered, for example, common formic acid, methanol, and ethanol. Further evaluation of the impact of the above

possible products on quantitative methods is needed to expand the scope of application of the conclusion.

Response:

Thanks for your comments. To verify whether the carbon-containing products (formic acid, methanol and ethanol) will affect the urease decomposition detection method, we prepared a series of urea standard solutions (concentration of 5 ppm) with addition of 50 mM formic acid, 50 mM methanol and 50 mM ethanol respectively. Then, these urea standard solutions with added carbon-containing products were decomposed by urease method, the obtained corresponding absorbance curve by UV spectrophotometer of these solutions with and without urease decomposition were compared and list in Figure R2. It is found that these UV absorbance curves before and after addition of carbon-containing products are basically coincident, indicating that the influence of carbon-containing products on the urease decomposition detection method is almost negligible.

Figure R2 | (a) The UV absorption curves comparison of standard urea solution with or without addition of 50mM formic acid before and after urease decomposition. (b) The UV absorption curves comparison of standard urea solution with or without addition of 50mM ethanol before and after urease decomposition. (c) The UV absorption curves comparison

of standard urea solution with or without addition of 50mM methanol before and after urease decomposition.

5. In the Supporting Information, “In site sum frequency generation” should be “In situ sum frequency generation”. In the Figure S5d and the Figure S6, the units for the abscissa are missing, which needed to be supplemented. The authors should also check the whole manuscript to correct some of typos.

Response:

Thanks for your comments. According to your suggestions, the whole manuscript has been carefully checked, the grammar and spelling mistakes have been thoroughly corrected.

Reviewer #3 (Remarks to the Author):

General comments for XANES: Except edge position drifting, other XANES features have been experimentally revealed by the reported XANES data, which is yet not fully addressed. Fe and Ni K edges XANES reported by SI Fig.6 (a) & (b) reveal well resolved XANES difference throughout the sample system in (1) pre-edge feature, (2) lineshape throughout the edge jump, (3) peak drifting, peak splitting, overall lineshape changing, and peak amplitude varying of XANES whiteline, and (4) significant difference in the post whiteline XANES region. These well resolved features are specific to the Fe and Ni local coordination environments, featured by their occupied sites, respectively. It is suggested to further address these experimentally revealed XANES features, correlating them to the local structural environment for the Fe and Ni occupied site predicted by DFT.

Response:

Thanks for your precious comments. XANES is kind of powerful tool to analysis materials' structure and reveal the coordination environments of active sites. We're sorry that we didn't process the XANES data professionally and analysis it well due to our lack of XANES knowledge. In the revised version, according to your suggestions, we have re-fit and further analyzed the XANES data to obtain more detailed structure information on the Fe and Ni local coordination environments to correlate them with DFT results well. Meanwhile, we will strengthen the study and understanding of XANES knowledge to better reveal the intrinsic active sites and catalytic mechanism of electrocatalysis resort to the powerful means.

Some other further comments:

1. Related contents: "peak position situates at lower energy than those of Fe foil..."in Manuscript, Page 3. Comments: A typo: "lower" should be replaced by "higher".

Response:

We are sorry for the mistake, and the "lower" have been revised to "higher" in the revised version. The following sentence of "the X-ray absorption near edge structure (XANES) data at the Fe K-edge of Fe-SAC, I-FeNi-DASC and B-FeNi-DASC clearly indicate that the peak

position situates at lower energy than those of Fe foil (Supplementary Fig. 6)” have been revised to “the X-ray absorption near edge structure (XANES) data at the Fe K-edge of Fe-SAC, I-FeNi-DASC and B-FeNi-DASC clearly indicate that the peak position situates at higher energy than those of Fe foil (Supplementary Fig. 6)” in the revised manuscript.

2. Related contents: X-ray absorption fine structure (FT-EXAFS) analysis

Comments: There is no analysis here, simply presenting magnitude of FT.

Response:

Thanks for your comments, we further analyzed the X-ray absorption fine structure (FT-EXAFS) and the following words “In terms of extended X-ray absorption fine structure (EXAFS) of the catalyst of B-FeNi-DASC, the predominant peaks appeared in the first coordinated shells (1-2 Å) upon R space curve of Fe K-edge and Ni K-edge, which originate from the scattering of 1st shell Fe-N and Ni-N path, are almost the same in position (at ~1.5 Å) and magnitude, indicating nearly identical coordination environment for Fe and Ni atoms in the catalyst of B-FeNi-DASC. Notably, the broad peak appears upon the 2nd shell of Fe K-edge and Ni K-edge with a single-scattering path of at around $2.49 \pm 0.07 \text{ \AA}$ appeared in the second scattering shells (2-3.5 Å) (Fig. 1d, Supplementary Fig. 7 and Supplementary Table 1). This distance is in the range of the observed separation of dual-atom pairs in atomic resolution STEM imaging (Fig. 1c), which is consistent with the previous reports,^[25,26] therefore, we attributed this scattering path to the formation of Ni-Fe dual-atom pairs, in which a Fe atom connect to the Ni atom except for coordinate with 3N sites. Taken together, Ni-Fe diatomic configuration is formed in the B-FeNi-DASC. By contrast, for the catalyst of I-FeNi-DASC, the first coordinated shells for both Fe and Ni are similar to that of B-FeNi-DASC, but the peak in the 2nd shell of EXAFS R space is gentle, indicating that there are almost no Fe-Ni pairs exist in the catalyst of I-FeNi-DASC. On the other hand, compared to Fe-SAC and Ni-SAC, the 1st shell scattering (Fe-N and Ni-N) for B-FeNi-DASC displays asymmetry and slightly decreased magnitude, indicating that the chemical state of Fe is altered by the coupling Ni atom. Wavelet transform (WT)-EXAFS was also conducted to identify the metal-N and metal-metal paths in B-FeNi-DASC” have been added in the manuscript.

The following literatures have been cited.

[25] Zeng, Z. et al. Orbital coupling of hetero-diatom nickel-iron site for bifunctional electrocatalysis of CO₂ reduction and oxygen evolution. *Nat. Commun.* **12**, 4088, (2021).

[26] Li, X. et al. Microenvironment modulation of single-atom catalysts and their roles in electrochemical energy conversion. *Sci. Adv.* **6**, eabb6833 (2020).

3. Related contents: peak at approximate 2.5 Å

Comments: Difficult to identify where peak is for B-FeNi-DASC, suggesting to arrow indicated.

Response:

According to your suggestion, the arrow has been added as present in Figure R3, and the corresponding figure has been revised in the manuscript.

Figure R3 | Fourier transform extended X-ray absorption fine structure (FT-EXAFS) spectra of Fe-SAC, I-FeNi-DASC and B-FeNi-DASC.

4. Related contents: proving the successful fabrication of isolated and bonded diatomic site configurations

Comments: EXAFS can only provide 1D info in terms of R and CN. Not info about the band angle is completely missing. Thus, it cannot fully prove the “site configurations”

Response:

Thank for your comments. Indeed, it is unreasonable on the expression of “site configuration” without three-dimensional structural information as support, therefore, we change the “site configuration” to “Fe-Ni pairs”. And the expression was also revised in XANES discussion section in the manuscript.

5. Related contents: Supplementary Figure 7 (SI, page 8)

Comments: SI Fig. 7(a) vs (b): Better quality in EXAFS R space curve fitting was obtained upon the 2nd shell between ~R2-3 Å for Ni K edge data vs that for Fe K edge regarding sample B-FeNi-DASC. Please address whether the difference in fitting quality between two fitting is induced by the defect of the DFT model, which was used to guide the reported fitting, either structurally and/or chemically, or induced by other reason(s)?

Response:

Thanks for your comments. After comprehensive analysis, we deduced that the main reason caused the difference in fitting quality upon the 2nd shell between ~R2-3 Å for Ni K edge and that for Fe K edge is induced by unsuitable parameter setting carried on the Fourier transform treatment for Fe k-edge. We re-exported the data of Fe k-edge through Fourier transform as shown in Supplementary Figure 7. The re-exported Fe K-edge exhibits improved fitting quality.

Supplementary Figure 7 | Experimental and fitted EXAFS spectra of the Fe K-edge in (a) B-FeNi-DASC, (b) Ni K-edge in B-FeNi-DASC. The insets show the corresponding fitted structural modeling. Purple, blue, navy blue and grey spheres refer to Fe, Ni, N and C atoms, respectively.

6. SI Fig.7(d): Data and Feff fit match reasonably well for the imaginary part of FT of Ni K edge EXAFS for the outer shell region $\sim 2.5-4.0$ Å, but there is well-resolved difference or the magnitude of FT between Data and Feff fit throughout this outer R region. Please address what is the reason for this inconsistency in fitting quality between the magnitude of FT and the imaginary part of FT?

Response:

Thanks for your comments. The inconsistency in fitting quality between the magnitude of FT and the imaginary part of FT is mainly due to the high data noise for the imaginary part of Ni K edge EXAFS at shell region $\sim 2.5-4.0$ Å, so as to the K-range parameter range setting is too narrow for Fourier transform, which leads to wide fluctuation range at shell region $\sim 2.5-4.0$ Å. Based on this, the data of Ni K edge EXAFS was conducted to reduce the noise to improve the fitting quality, the revised figure is shown in Supplementary Figure 7.

Supplementary Figure 7 | (c) Fe *K*-edge in I-FeNi-DASC, (d) Ni *K*-edge in I-FeNi-DASC, (e) Fe *K*-edge in Fe-SAC and (f) Ni *K*-edge in Ni-SAC. The insets show the corresponding fitted structural modeling. Purple, blue, navy blue and grey spheres refer to Fe, Ni, N and C atoms, respectively.

7. What is the impact of the revealed fitting defect to those fitted structure parameters for the corresponding outer shell Ni coordination (SI table 1)? The accuracy for which fitted parameter(s) is impacted, and in what scale? Fig. 7(e): certain fitting defect is revealed for

Fe outer shell fit for the imaginary part of FT for region $\sim R(2.5-3.0 \text{ \AA})$. Please clarify whether this is a structure effect or chemistry effect?

Response:

Thanks for your comments. The fitting defect may produce certain effect on some fitted parameters for outer shell Ni coordination, such as bond length (interatomic distance), the coordination number, type of coordination atoms and Debye-waller factor. The proportion of these parameters that may be affected are listed as follow: Bond length ($\pm 1\%$), the coordination number ($\pm 20\%$), type of coordination atoms (± 4) and Debye-waller factor ($\pm 20\%$). For Supplementary Figure 7e, we deduced that the fitting defect for Fe outer shell is caused by unsuitable parameter setting carried on Fourier transform and fitting. Based on this, we have re-exported and fitted the data of Fe k-edge through Fourier transform as follow.

Supplementary Figure 7 | Experimental and fitted EXAFS spectra of the Fe K-edge in (e) Fe K-edge in Fe-SAC.

8. Error bars are missing for all reported parameters.

Response:

Thanks for your comments. The error bars of EXAFS curve-fitting parameters have been added as follow.

Supplementary Table 1 | Fe and Ni K-edge EXAFS curve-fitting parameters for SACs and reference samples.

Samples	Path	N	R(Å)	$\sigma^2(\times 10^{-3} \text{ \AA}^2)$	ΔE_0	R-factor
Fe-foil	Fe-Fe	7.8±0.4	2.46±0.002	5.0±0.46	5.5±1.9	0.02
Ni-foil	Ni-Ni	12±0.5	2.47±0.002	5.0±0.62	6.4±1.3	0.0176
B-FeNi-DASC	Fe-Ni	5.8±0.4	2.49±0.001	8.0±0.34	6.2±1.2	0.016
I-FeNi-DASC	Fe-N	3.8±0.7	2.04±0.001	8.0±0.56	5.3±1.3	0.04
I-FeNi-DASC	Ni-N	3.8±0.4	1.99±0.002	7.0±0.75	5.3±2.0	0.04
Fe-SAC	Fe-N	3.9±0.2	2.09±0.001	6.0±0.63	4.2±2.2	0.033
Ni-SAC	Ni-N	3.8±0.6	2.10±0.001	8.1±0.49	6.9±2.5	0.04

Note: N is coordination number, R is the distance between absorber and backscatter atoms, σ^2 is Debye-Waller Factor, E_{shift} is inner potential correction; R -factor indicates the goodness of the fit.

9. If coordination numbers (N) are all fixed throughout fitting for all scattering paths, either fixed upon crystallography of metallic Fe/Ni, and/or to what DFT predicted, please specify in the table.

Response:

Thanks for your comments. The coordination numbers (N) are not fixed throughout fitting process, the obtained coordination numbers (N) are average values over the whole system based on the constructed models predicted by DFT, and we take integer of these values according to the crystal structure information of Ni and Fe in this work. The original values of the obtained coordination numbers (N) with error bar were list in Supplementary Table 1 in the revised manuscript.

Reviewer #4 (Remarks to the Author):

The electrocatalytic C-N coupling serves as a promising alternative to the traditional urea synthesis and the emerging field of electrocatalysis research. In this work, the authors designed a diatomic catalyst with bonded Fe-Ni pairs to improve the efficiency of electrochemical urea synthesis. The “three-in-one” of active site, activation site and coupling site was realized in the FeNi-N₆ sites. The origin of the superior electrocatalytic activity for FeNi-N₆ was experimentally and theoretically confirmed. This innovative research work achieved the identification and tailoring of C-N coupling sites for electrocatalytic urea synthesis, which is expected to promote the green revolution of urea industrial and provide broaden guidance for catalytic coupling reactions. Considering the significance of this work in the related fields, I would like to suggest its publication in Nature Communications after a minor revision.

Thanks for your approval to this manuscript. The information of the materials has been supplemented and the negligible leaching of metals during electrochemical measurements has been verified.

1. The information of the materials used must be afforded in the section of methods, including the purity of carbon dioxide gas, the manufacturer and purity of chemical reagents.

Response:

Thanks for your comments. The following words “Chemicals. Carbon papers (HCP020N) were purchased from Hesen Company (Shanghai, China) and washed with water and acetone before use. KHCO₃ (≥99.99% metals basis, 99.7-100.5% dry basis) was bought from Aladdin Biochemical Technology Co., Ltd. KNO₃ (>99%), were received from Sinopharm Chemical Reagent Co., Ltd. Urease from Canavalia ensiformis (Jack bean) was purchased from Sigma, Urease activity: 20KU, Batch number: Lot#SLCJ5647. Ar (>99.999%), N₂ (>99.999%), CO₂ (>99.999%) were bought from Changsha Rizhen Gas Co., Ltd. All chemicals were used without further purification.” have been added in the section of Methods in the revised manuscript.

2. The urease decomposition method was adopted in this work for the quantification of urea product. The UV-vis adsorption curves for the urea quantification need to be complemented in the Supplementary Information.

Response:

Thanks for your comments. The UV-vis adsorption curves for the urea quantification are presented as followed and the results have been supplemented in the revised Supplementary Information as Supplementary Figure 16.

The following words “The corresponding absorbance curve by UV-vis spectrophotometer of B-FeNi-DASC under the applied potential of -1.4 V versus RHE with and without urease decomposition were illustrated in Supplementary Figure 16” are supplemented.

Supplementary Figure 16 | The corresponding absorbance curve by UV-vis spectrophotometer with and without urease decomposition. The absorbance at 662 nm was measured by UV-vis spectrophotometer. The insets are the optical images for color generation assays.

3. The structural stability is highly important for the recognition of active sites and the evaluation of catalyst stability. The author needs to confirm whether there is metal leaching under the working condition.

Response:

Thanks for your comments. In order to confirm whether there is metal dissolution during the process of electrochemical synthesis of urea, the electrolyte after 5 h continuous test

was extracted and the metal content was measured by Inductive Coupled Plasma Emission Spectrometer (ICP). The test protocol is presented as followed: 5 mL electrolyte mixed with 5 mL HNO₃ solution (8 M), and then the mixed solution was transferred to hydrothermal reactor under 180 °C for 24 h. After filtered by filter paper with a diameter of 0.22 μm, the obtained filtrate was conducted to detect the ion concentration of Fe and Ni leaching. The concentration of metals is lower than the detection line due to the ultra-low content. The results indicate that there is barely leaching of metal ions into the electrolyte after long-term measurement, implying the superior electrochemical stability of B-FeNi-DASC.

4. The quality of Figures needs to be further improved, such as Figure 1c and Figure 1d.

Response:

Thanks for your comments. The quality of figures has been improved and the revised Figures have been added in revised manuscript as followed.

Figure 1 | Morphology and structure of single-atom and diatomic catalysts.

5. There are some errors displayed in Figure 2b. The authors should check this point and the expressions in the manuscript.

Response:

Thanks for your comments. We have corrected Figure 2b and revised in manuscript. And

we have carefully checked the expressions in the whole manuscript.

Figure 2 | Electrocatalytic performances for urea synthesis. **b**, The product distributions of CO₂RR, NO₃RR and urea synthesis on Ni-SAC, Fe-SAC, I-FeNi-DASC and B-FeNi-DASC at -1.4 V versus RHE.

We would like to express our sincere thanks to the reviewers and editorial team of *Nature Communications*, and we think the quality of this manuscript has been significantly promoted attributing to the instructive comments.

REVIEWERS' COMMENTS

Reviewer #1 (Remarks to the Author):

The authors have carefully answered all my queries and am now happy to recommend the manuscript for publication

Reviewer #2 (Remarks to the Author):

The authors have addressed the comments raised by this reviewer. The reviewer would like to recommend the acceptance of the manuscript.